# Phytochemistry and Biological Activity of Medicinal Plants in Wound Healing: An Overview of Current Research

**DOI:** 10.3390/molecules27113566

**Published:** 2022-06-01

**Authors:** Stefania Vitale, Sara Colanero, Martina Placidi, Giovanna Di Emidio, Carla Tatone, Fernanda Amicarelli, Anna Maria D’Alessandro

**Affiliations:** 1Department of Life, Health and Environmental Sciences, University of L’Aquila, 67100 L’Aquila, Italy; stefania.vitale@guest.univaq.it (S.V.); martina.placidi@graduate.univaq.it (M.P.); giovanna.diemidio@univaq.it (G.D.E.); carla.tatone@univaq.it (C.T.); fernanda.amicarelli@univaq.it (F.A.); 2Department of Biosciences, University of Milan, Via Giovanni Celoria 26, 20133 Milan, Italy; sara.colanero@unimi.it

**Keywords:** wound healing, herbs, medicinal plants, secondary metabolites, natural wound dressing, skin disorders

## Abstract

Wound healing is a complicated process, and the effective management of wounds is a major challenge. Natural herbal remedies have now become fundamental for the management of skin disorders and the treatment of skin infections due to the side effects of modern medicine and lower price for herbal products. The aim of the present study is to summarize the most recent in vitro, in vivo, and clinical studies on major herbal preparations, their phytochemical constituents, and new formulations for wound management. Research reveals that several herbal medicaments have marked activity in the management of wounds and that this activity is ascribed to flavonoids, alkaloids, saponins, and phenolic compounds. These phytochemicals can act at different stages of the process by means of various mechanisms, including anti-inflammatory, antimicrobial, antioxidant, collagen synthesis stimulating, cell proliferation, and angiogenic effects. The application of natural compounds using nanotechnology systems may provide significant improvement in the efficacy of wound treatments. Increasing the clinical use of these therapies would require safety assessment in clinical trials.

## 1. Introduction

The skin plays a vital role in protecting the body from the external environment. Damage to skin integrity caused by incisions, burns, scalds, and human lesions (diabetic foot, venous ulcers, pressure sores, etc.) is considered a wound. Effective treatment of wounds is a major challenge for human health. Failure to heal and a prolonged healing time increase the economic and social costs for healthcare institutions and professionals, patients, and their families [1,2,3]. Improper treatment of wounds may result in many issues, e.g., bleeding, infection, inflammation, scarring, and difficulty with angiogenesis and regeneration [4]. The World Health Organization (WHO) estimates that approximately 80% of the world’s population depends on traditional practices for primary health care, and 85% of this portion uses plants. Considering the difficult access of these populations to conventional forms of treatment, the WHO suggests the adoption of traditional practices as a tool for maintaining health and encourages the development of public policies to insert them into the official health system of its 191 member countries. According to WHO statistics, approximately 5 million people could die each year because their wounds do not heal properly.

Phytochemicals have shown significant promise in the prevention and treatment of microbial infections and wounds [5,6]. Anti-microbial, antioxidant, and wound healing phytochemicals encourage blood clotting, fight infection, and accelerate the healing of wounds. Medicinal plants rich in polyphenols are reported to possess remarkable wound healing activity [7,8,9]. Phenolics promote wound healing mainly due to their astringent, antimicrobial, and free radical scavenging properties [10,11]. Finally, polyphenolic components such as flavonoids can promote excellent healing of wounds, likely by means of antimicrobial and anti-oxidative properties, by inhibiting lipid peroxidation, which leads to the prevention of cell damage and the increase in the viability of collagen fibrils [12,13]. Many people in the less-developed parts of the world still use traditional medicine, especially for treating wounds [14].

Recently, a growing number of research articles have been focusing on the employment of herbal natural products as potential beneficial agents in the process of wound healing. Indeed, the low cost, high availability, and reduced side effects are the principal advantages of these botanical remedies. In plants, a broad spectrum of bioactive phytochemicals is present, mainly belonging to the families of alkaloids, carotenoids, flavonoids, tannins, terpenoids, saponins, and phenolic compounds [15]. This article gives a general overview of the recent advances in traditional therapies for skin wound healing, focusing on the therapeutic activity, action mechanisms, and clinical trials of the most commonly used natural compounds. New information on how traditional products can be used with modern treatments and how the field will change in the future is also discussed.

## 2. Methods

The literature on wound biology, healing, and the role of herbs in wound healing was searched in the science databases PubMed, Scopus, and Cochrane Library. Only peer-reviewed papers with authenticated references and sufficient data as evidence were considered. In vitro, in vivo, and clinical studies were considered in the literature collection on wound biology, healing, and the role of herbs in wound healing. The search terms were “wound healing” in the title and abstract, and “plant,” “extract,” or “herb” in the whole text. According to this first screening, approximately 450 plant species with wound-healing properties have been identified, including numerous review articles.

In order to make an original contribution to the field in this article, we have considered it relevant to focus on research articles not yet included in previous review articles. To this end, plants, molecules, and effects on wound healing investigated in the last five years have been specifically highlighted and thoroughly discussed. So, this review sums up the new information on the possible benefits of using natural medicines and the underlying mechanisms.

## 3. Wound Healing Process

Wound healing is characterized by a well-orchestrated series of events aimed at ultimately restoring the barrier function and mechanical integrity of the skin [16]. Tissue damage starts the healing process, which is made up of four time-dependent stages: Coagulation and hemostasis, inflammation, proliferation, and wound remodeling (Figure 1) [17].

### 3.1. Hemostasis

Immediately after injury (Figure 1A), coagulation and hemostasis take place to prevent bleeding [18,19]. These processes require multiple interlinked steps to protect the blood vessel and provide a matrix that allows the invasion of cells needed in the later phases of healing [20,21]. The coagulation cascade is activated through extrinsic and intrinsic pathways, leading to platelet aggregation and clot formation [19]. In rapid succession, three steps occur during hemostasis. Vascular spasm is the initial response as blood vessels constrict to reduce blood loss. In the second step, within seconds of a blood vessel’s epithelial wall being disrupted, platelets adhere to one another to form a temporary seal over the vessel wall rupture. Collagen, exposed at the site of injury, promotes platelets to adhere to the injury site. The third and final phase is known as blood clotting or coagulation. The coagulation cascade, or secondary hemostasis, is a series of steps in response to bleeding caused by tissue injury, where each step activates the next and ultimately produces a blood clot. Coagulation strengthens the platelet plug with fibrin threads, which function as “molecular glue.” It takes approximately 60 s for the first fibrin strands to begin to intertwine with the wound. After a few minutes, the fibrinous platelet block is entirely established [19,20,21,22]. Platelet-derived growth factors and cytokines also play a role in the wound-healing process by activating and attracting neutrophils, as well as macrophages, endothelial cells, and fibroblasts [17,21].

### 3.2. Inflammatory Phase

This phase is composed of two separate steps, an early inflammatory phase and a late inflammatory phase (Figure 1B) [23]. The early inflammatory phase activates the molecular events that lead to the infiltration of the wound site by neutrophils, whose main function is to prevent infection [24]. The neutrophils begin to be attracted to the wound site within 24–36 h of injury by various chemoattractive agents. Once in the wound environment, neutrophils remove foreign material, bacteria, and damaged tissue by releasing proteolytic enzymes and oxygen-derived free radical species [17,22,25]. Upon completing the task, the neutrophils are eliminated from the wound. During the late inflammatory phase, macrophages appear in the wound and continue the process of phagocytosis before progressing to the next phase of healing [23,24,26,27,28,29,30,31]. Macrophages play an important role in the late stages of the inflammatory response, acting as key regulatory cells and storing a large amount of potent tissue growth factors [17,24,27,28,29,30,31,32,33]. The last cells to enter the wound site (in the late inflammatory phase) are lymphocytes, which play an important role in collagenase regulation and are later required for collagen remodeling, the production of extracellular matrix components, and their degradation [23,29,31].

### 3.3. Proliferative Phase

This phase encompasses the major healing processes and starts on the third day after wounding and lasts for approximately 2 weeks (Figure 1C). Following the injury, fibroblasts and myofibroblasts proliferate in the surrounding tissue for the first three days [30] before migrating into the wound, attracted by factors such as Transforming Growth Factor- (TGF-) and Platelet-Derived Growth Factor (PDGF), which are released by inflammatory cells and platelets [34]. Once in the wound, the fibroblasts proliferate profusely and proceed to the deposition of the newly synthesized extracellular matrix by producing matrix proteins such as hyaluronan, fibronectin, proteoglycans, and type 1 and type 3 procollagen. Collagen synthesis represents a crucial event. Indeed, collagen is an important component in all phases of wound healing since it contributes to the integrity and strength of all tissues [35,36,37]. At a macroscopic level, during this phase of wound healing, an abundant formation of granulation tissue can be observed. Then, within a few days, a microvascular network throughout the granulation tissue is organized. The modeling and establishment of new blood vessels are critical in wound healing and take place concurrently during all phases of the reparative process. Finally, epithelialization occurs. Within a few hours of the injury, epithelial cells start to migrate to the wound site from the wound edges; when the advancing epithelial cells meet, migration stops, and the basement membrane starts to form [29].

### 3.4. Remodeling Phase

During wound remodeling (Figure 1D), scar tissue formation takes place and is completed in approximately one year or even more [18,29,38,39,40]. As the wound heals, the growth of capillaries stops, the density of fibroblasts and macrophages is reduced by apoptosis, and the blood flow and metabolic activity decrease [35,37,41]. The final result is a fully matured scar with a decreased number of cells and blood vessels and high tensile strength [23,41]. Although most wounds are the result of simple injuries, systemic and local factors may alter and slow down the course of the finely balanced repair process, leading to the occurrence of wounds that do not heal in a timely and orderly manner and evolve in chronic, non-healing wounds. Therefore, according to the tissue’s ability to completely repair the wound, they are classified generally as acute or chronic wounds.

## 4. Acute and Chronic Wounds

While acute wounds take only a few weeks to heal, chronic wounds require several months to heal completely. An acute wound is defined as a recent wound that has yet to progress through the sequential stages of wound healing. They can be superficial injuries involving both the epidermis and superficial dermis or full-thickness skin damage in which the subcutaneous layer is compromised. Examples of acute wounds are surgical incisions, thermal wounds, abrasions, and lacerations. They heal through the routine processes of inflammation, tissue formation, and remodeling, which occur in a timely fashion. Acute wound healing is regulated by cytokines and growth factors released proximal to the wound [42].

Chronic wounds are those that fail to progress through the normal stages of healing and are not repaired in an orderly and timely manner [20,43]. The healing process is incomplete and disturbed by various factors, which prolong the phases of homeostasis, inflammation, proliferation, or remodeling by one or more stages. These factors can include infection, tissue hypoxia, necrosis, exudate, and excess levels of inflammatory cytokines [38]. The continuous state of inflammation in the wound creates a cascade of tissue responses that together perpetuate a non-healing state. Since the healing proceeds in an uncoordinated manner and the functional and anatomical outcomes are poor, these wounds frequently relapse [20,39]. Chronic wounds may be related to various causes, including neuropathic diseases, pressure, arterial and venous insufficiency, burns, and/or vasculitis (Figure 2) [44].

## 5. Dressing Wounds

Medical dressings are essential devices in healthcare and are an essential element of standard wound care. The therapeutic effects of traditional dry dressings and modern wet dressings in the clinical management of wounds are both well documented [45]. According to the types and stages of wounds, dressings can be applied to their surface to promote healing. Even though common clinical practice dressings (gauze and bandages) are inexpensive, they only provide physical protection and have limited benefits in terms of the healing process and infection prevention. Moreover, adherence of the dressing to the wound can cause secondary damage when the two are eventually separated [45]. Wound dressings with high performance should be capable of keeping the wound moist, ensuring the solubilization of growth factors, supporting fibroblast growth, as well as absorbing exudate and exchanging oxygen. Furthermore, the ideal wound dressing should also have antibacterial activity and biocompatible properties [46].

## 6. Natural Herbals and Spices in Wound Healing

Herbal medicines have been employed in folk medicine to accelerate wound healing since ancient times. Many plants and various preparations thereof have been used traditionally in relation to wound treatment, especially due to their immense potential to affect wound healing [47]. Plant-based extracts and/or isolates support tissue regeneration through a variety of mechanisms, which often work together to improve the whole healing process [48].

Currently, the efficacy of many of these herbs is well documented together with their mechanisms of action [49]. Therefore, natural products as well as their pure compounds are an emerging source of alternative medicinal compounds for the management of various diseases, among which is wound healing [8,50,51].

In recent years, extensive research has been carried out in the area of wound healing and management through plant-derived medicinal products [48]. The herbs presented here were chosen because they are extensively used in wound healing. The following subchapters will review the more recent papers, published in the last five years, related to the potential use of medicinal plants in wound healing.

Figure 3 give a summary of the major plant-derived phytochemicals that successfully contribute to the alleviation of the inflammatory state and healing process. 

### 6.1. Achillea millefolium

*Achillea millefolium* L. (yarrow) is an important species of *Asteraceae* family with common utilization in the traditional medicine of several cultures from Europe to Asia for the treatment of various disorders including wounds, cuts, abrasions, and diabetic ulcers [52,53,54,55]. Essential oils, the most active part of the yarrow flower, are used in wound therapy as a hemostatic [56,57,58]. The most representative metabolites in volatile oil extracted from *Achillea* flowers are monoterpenes (90% of the essential oils). However, there are reports on the presence of higher levels of sesquiterpenes compared with monoterpenes [59,60,61]. A wide range of other chemical compounds has also been reported in the methanolic extract of *A. millefolium* aerial parts (Figure 3). Among them, flavonoids such as chlorogenic acid, rutin, luteolin-glucoside, and apigenin-glucoside account for over 90% of the total extract components [62]. In different in vitro and in vivo experiments, active compounds derived from *A. millefolium* showed antioxidant and anti-inflammatory, mostly attributed to the presence of flavonoids [55,63,64,65,66,67,68]. Despite yarrow being extensively used for wound healing and skin inflammatory disorders in traditional medicine [68], modern studies of yarrow extract effects on wound healing are lacking.

In a recent paper, Dorjsembe et al. [69] investigated the effect of the alcoholic extract of *Achillea asiatica*
*flowers* on cutaneous wound healing, in in vitro and in vivo models, showing that 3% YE treatment significantly increased epithelialization and accelerated wound healing in a rat model. This effect was associated with an increase in β-catenin, and Akt expressions [69]. Furthermore, in in vitro experiments, YE treatments (25–50 mg/mL) stimulated collagen expression by activating transforming growth factor-β (TGF-β) in Hs68 fibroblasts and reduced Nitric Oxide (NO) and prostaglandin E2 released, in RAW 264.7 macrophages, reflecting anti-inflammatory activity. Based on these results, a beneficial positive effect of yarrow flower extract on wound healing can be hypothesized [69].

### 6.2. Aloe vera

*Aloe vera* L. or *Aloe barbadensis Miller* (family *Xanthorrhoeaceae*) is a perennial green herb with bright yellow tubular flowers. The mucilaginous gel has been extensively used in pharmacological and cosmetic applications [70]. It has more than 75 different compounds, including vitamins A, C, E, and B12, enzymes (amylase, catalase, and peroxidase), minerals (zinc, copper, selenium, and calcium), sugars (glucomannans, acetylated mannans, polymannans), polyphenols (anthraquinones), sterols (lupeol and campesterol), and hormones (auxins and gibberellins) [71,72,73]. Traditionally, the therapeutic uses of *A. vera* have ranged across a broad list of conditions, as do its associated pharmacological activities. This medicinal plant has been employed to treat different skin problems such as rejuvenation, wound healing, and other dermatologic conditions, i.e., burns and inflammatory processes. Indeed, despite its widespread use as a folk remedy, scientific studies on its physiological function in wound repair have only recently been conducted [72]. Numerous *A. vera* gel-based cosmetics and medicinal products that are marketed are made from the mucilaginous tissue of *A. vera* leaves. However, Aloe gel has been linked to a variety of beneficial effects and therapeutic indications for skin inflammation [72].

Individual components of *A. vera* gel may promote wound healing in animal models, and specific glycoproteins are responsible for the beneficial effects when the gel is applied to acute wounds in various animal models; however, controlled clinical trials in humans demonstrated no benefit when *A. vera* was incorporated into topical therapy, and one study reported delayed wound healing [74,75,76]. Recently, topical use of *A. vera* in rat cutaneous wound models has shown that it significantly increases the rate of wound contraction, epithelialization, and maturation. Moreover, it reduces inflammation, decreases scar tissue size, and increases alignment and organization of regenerated scar tissue, increasing the concentration of collagen and glycosaminoglycans compared to the control lesions [77].

To clarify the mechanism by which *A. vera* modulates inflammation, Hormozi and colleagues [78] exposed mouse embryonic fibroblast cells to different concentrations (50–150 µg/mL) of *A. vera* gel for 12 and 24 h [78] and found that Transforming growth factor -beta1 (TGF1) and Fibroblast Growth Factor basic Protein (bFGF) genes were found upregulated during the first 12 h and down-regulated at the end of incubation. This suggests that *A. vera* exerts beneficial effects by stimulating collagen deposition, fibroblast proliferation, and angiogenesis and inhibiting the overproduction and accumulation of matrix proteins that cause hypertrophic scarring [78].

In in vitro models of human skin fibroblasts and keratinocytes [79,80,81,82], *A. vera* accelerated wound healing by strongly stimulating fibroblast and keratinocyte proliferation and moderately stimulating cell migration. Surprisingly, *A. vera* also protected keratinocytes against preservative-induced death [83]. These previously unknown protective actions may account for some of the beneficial benefits of *A. vera* in wound healing.

According to a recent review *A. vera* compounds such as aloesin can modulate the inflammatory response [82]. Aloesin promotes leukocyte extravasation as well as cytokine and growth factor release during the early stages of wound healing. As a result, aloesin appears to influence both fibroblast and leukocyte migration [78]. Aloesin is thought to influence leukocyte migration by phosphorylating Cdc42 and Rac1, two signaling proteins that coordinate and regulate actin dynamics and cell polarization [79]. In the presence of aloesin, TNF-alpha, Interleukin 1 beta (IL-1beta), Interleukin 6 (IL-6), and TGF-1 are pro-inflammatory markers that mediate leukocyte signaling, migration, and phagocytosis [80]. Recently, a study by De Oliveira et al. [81] established that the healing property of *A. vera* extract can be observed at a dose of 50 mg/kg, but at the same dose, mutagenic and cytotoxic effects are observed in peripheral blood [81].

Aside from flavonoids such as aloin and emodin, polysaccharides such as glucomannan, acetylated polymannan, acemannan, and mannose-6-phosphate appear to promote wound healing [73] (Figure 3). Some studies showed that acemannan appears to stimulate macrophages and boost bactericidal activity [75,76], the expression of keratinocyte growth factor-1 (KGF-1), cell proliferation, as well as vascular endothelial growth factor (VEGF), an important molecule in the formation of new blood vessels, and type I collagen synthesis [84,85]. Studies in skin fibroblasts revealed that acemannan not only increased proliferation but also shifted the cell cycle from the G1 to S phase, increasing the expression of proliferation markers such as cyclin D1 in a dose-dependent manner [86,87].

Some studies have found no significant differences between *A. vera* treatment and control groups in terms of wound healing, with little evidence of decreasing microorganism concentrations or improving scar maturation in burn wounds, no significance in fibroblast migration dynamics, and no difference between sample groups in terms of wound repair [82]. This is likely due to a problem of standardization of extract concentrations. Some of these studies may have used aloe extracts with insufficient concentrations of active compounds with known anti-inflammatory and proliferative effects, or they may have been combined with other unsuitable alternative therapies.

### 6.3. Bletilla striata

*Bletilla striata* is a member of the *Orchidaceae* family and has been used in Traditional Chinese Medicine for over 1500 years to promote wound healing and treat alimentary canal mucosal damage, chapped skin, ulcers, bleeding, bruises, and burns [88]. Phytochemical research on *B. striata* yielded the identification of 192 monomeric compounds. These compounds, extracted from *B. striata* tubers, are primarily triterpenoids: Phenanthrenes, biphenanthrenes, dihydrophenanthrenes, anthocyanins, quinones, steroids, glucosides, bibenzyls, and phenolic acids [89]. According to pharmacology studies, the plant has many biological activities, including antioxidative, anti-inflammatory, and immune regulatory effects [89]. *Bletilla striata* also contains several polysaccharides, which have been identified as the major active components in its dried tubers, and are responsible for antimicrobial, anti-aging, antioxidative, and antiviral activity [90]. Natural polysaccharides have been used in a variety of biomaterials in recent years due to their high biocompatibility, low toxicity, and pharmaceutical biomedical activity. *Bletilla striata* polysaccharides (BSP) can be used as natural biomaterials for drug delivery and wound dressing in addition to promoting wound healing [91]. The most common polysaccharide in BSP is glucomannan, which is made up of D-glucose and D-mannose and has a high molecular weight [92]. BSP plays a critical role in the three main phases of wound healing: Inflammation, granulation tissue proliferation, and repair [93]. During the inflammation phase, BSP promotes the expression of inflammatory mediators such as tumor necrosis factor (TNF)-, interleukin (IL)-1, and interferon (IFN)-, increases NO, and promotes neutrophil, monocyte, and macrophage chemotaxis [94]. By controlling the expression of TNF-, BSP can reduce the inflammatory reaction in the wound and prevent damage to remaining cells during the phases of granulation tissue proliferation and repair [95]. These actions promote epithelial cell growth, fibroblast proliferation, and wound healing by contracting the wound [88,96]. Yue and colleagues [97] measured the response induced by BSP pretreatment in terms of ROS levels and proinflammatory cytokines after Ang II stimulation with and without BSP and investigated the signaling pathways potentially involved in the anti-oxidative stress and anti-inflammatory functions mediated by BSP in a human mesangial cell model in vitro (HMCs) study. BSP was found to inhibit the generation of Ang II-induced reactive oxygen species (ROS) as well as the activation of pro-inflammatory cytokines such as interleukin 6 (IL-6) and tumor necrosis factor (TNF), in a dose-dependent manner (5–80 µg/mL). Furthermore, BSP effectively inhibited NADPH oxidase 4 upregulation (NOX4). Indeed, NOX4 knockdown significantly reduced Ang II-induced TLR2 overexpression and blocked TLR2 expression, impairing BSP’s anti-inflammatory effects. Furthermore, BSP has anti-oxidative and anti-inflammatory properties that can help stop ROS production and the production of proinflammatory cytokines [97].

BSP derived from *B. striata* has been used in wound dressings for many years, with outstanding biocompatibility, biodegradability, and gelling properties [95]. BSP hydrogel demonstrated excellent performance in the absorption of wound fluids and exudates and was able to provide water to the wound, thus creating a suitable fluid environment for the wound bed [95]. Surprisingly, recent research has revealed that BSP has hemostatic activity, likely mediating blood clotting and fibrinolysis [98,99,100]. BSP has been made into a new type of hemostatic agent that can be used as a drug delivery vehicle and wound dressing. [75].

Ding et al. [101] created a chemically cross-linked composite bilayer as a spongy wound dressing and tested it on dermal wounds in mice. The upper layer was made of chitosan-Ag nanoparticles that were cross-linked with genipin, extracted from gardenia fruits [102], and exerted antibacterial activity. The lower layer was a hybrid cross-linked chitosan with genipin and partially oxidized BSP that revealed potent cell proliferation activities. Based on its ability to accelerate the healing rate of cutaneous wounds, this novel composite bilayer has a great deal of potential in wound-dressing applications [103]. Y. Song and colleagues [103] discovered that the fibrous residues of polysaccharide extraction from *B. striata* used in traditional medicine have a high phenolic content, strong DPPH (2,2-diphenyl-1-picrylhydrazyl) scavenging activity, ferric-reducing antioxidant capacity, and tyrosinase inhibition activity [103]. After 15 days of treatment, the wound shrinkage percent in mice treated with mix ointment containing 1% residue was statistically higher than in animals treated with saline solution, Bletilla phenolic ointment, or partially oxidized BSP ointment. Based on these findings, the authors concluded that wound-healing functions of *B. striata* are due not only to the composition of the polysaccharides, but also to the activity of the polyphenols [103].

In a more recent study, Cheng et al. [104] combined BSP and the ethanol extract of *B. striata* (EEB) to create new composite sponges through simple mixing and freeze-drying processes. When EEB (25–50% in H_2_O) was added to the BSP sponge, it influenced blood coagulation factors, shortened hemostasis time, and improved antiseptic properties for wound healing. The degradation, biocompatibility, hemostatic, bacteriostatic, and wound-healing-promotion properties were also tested in vivo and in vitro [104]. This plant could be used to make plant-based components for wound dressing and drug delivery systems on a large scale.

### 6.4. Calendula officinalis

*Calendula officinalis*, belonging to the *Asteraceae* family, is a common garden plant used medicinally in Europe, China, the United States, and India. It has numerous common names in use, including Marigold and Pot Marigold. Traditionally, it has been used externally to treat small wounds, burns, and other skin problems [105,106]. *C. officinalis* can be used in the form of an infusion, tincture, liquid extract, cream, or ointment for numerous uses, including the treatment of herpes, wounds, scars, and skin and hair products [105,106,107,108].

Over the past decade, chemical and pharmacological studies have found that *C. officinalis* possesses many secondary metabolites with various pharmacological properties that contribute to its medicinal use [107,108,109,110,111,112,113]. The most active components are triterpenoids, both in their free and esterified forms, flavonoids, coumarines, quinones, volatile oil, carotenoids, polyunsaturated fatty acids, such as calendic acid, and amino acids [82,111] (Figure 3).

Most of the research on the role of *C. officinalis* in healing acute wounds came from in vitro (fibroblasts and keratinocytes cells) and in vivo (rodent animal models) studies [110,113,114,115,116,117,118]. Clinical studies have focused on chronic wounds [111], and only one clinical trial looked at acute wounds [112]. *Calendula* flower alcohol extracts (CFE) have shown anti-inflammatory and antiedematous properties in vitro. Furthermore, they increased human fibroblast and keratinocyte proliferation and migration, promoted angiogenesis in the chorioallantois membrane model, and decreased collagenase activity [117]. CFE influenced the inflammatory phase in keratinocytes by activating the transcription factor NF-kB and increasing the amount of chemokine IL-8 at both transcriptional and translational levels, whereas it had a minor effect on keratinocyte migration. CFE induces fibroblast proliferation [110,118] and upregulates the expression of connective tissue growth factor (CTGF) and alpha-smooth muscle actin (alpha-SMA) during the early stages of wound healing [118]. In a previous in vivo study, CFE reduced the presence of fibrin and hyperemia while increasing collagen deposition in a rat cutaneous wound model. These findings indicate that CFE has anti-bacterial properties, stimulates fibroplasia and angiogenesis, and positively influences the inflammatory and proliferative phases of the healing process of cutaneous wounds [118]. Terpenoids seem to be primarily responsible for all these effects.

Recently, ethanolic CFE was loaded onto chitosan nanofibers as a wound-healing dressing [119], and potent antibacterial properties were observed, with more than 90% reduction of Gram-positive and Gram-negative bacteria. In vitro studies with fibroblast cells demonstrated this mixed film increased cell proliferation, growth, and adhesion and inhibited cell collagenase activity with a subsequent increase in collagen production. In vivo studies of rat wounds revealed that this dressing has an excellent wound-healing capacity (87.5% wound closure after 14 days) by enhancing collagen synthesis, re-epithelialization, and tissue remodeling [119]. In all the studies examined, CFE was associated with a statistically significant improvement in wound healing according to the outcomes measured [110,116,117,118,119].

### 6.5. Casearia sylvestris

*Casearia sylvestris* is a Brazilian herb in the *Salicaceae* family [120]. *C. sylvestris*, popularly named “guaçatonga”, is used topically to treat gastric ulcers, wounds, or promote cutaneous healing in different communities in Brazil. *C. sylvestris* contains many chemical constituents in the leaves [120]. These compounds, which primarily consist of volatile oils, tannins, and triterpene content [121], are responsible for a wide range of biological activities, including antioxidant and cytotoxic effects [122,123,124]. Pharmacological studies have shown that *C. sylvestris* extracts (ethanolic, hydroethanolic, and aqueous extracts) and bioactive compounds have antitumoral, antiulcerogenic, anti-inflammatory, and wound-healing properties [120,121,122]. The anti-inflammatory and wound healing actions have been attributed to the casearin-like clerodane diterpenes isolated from *C. sylvestris* leaves, such as ellagic acid [124,125,126]. Regarding the toxicity, *C. sylvestris* extracts and isolated diterpenes showed low toxicity both in vitro and in vivo [120,122,125,126,127,128,129,130].

De Campos et al. [128], using a burn wound model in rats, carried out daily wound treatments with a biofilm impregnated with *C. sylvestris* extract and observed significant extension of the healing area, neovascularization, fibroblast proliferation, and epithelialization, concluding that *C. sylvestris* may have a potential therapeutic benefit in second-degree scald burn injuries. This effect has been ascribed to rutin, a *C. sylvestris* bioactive compound with powerful antioxidant properties [122]. When *Casearia sylvestris* leaves extract was tested on wound healing in beef cattle, cattle, a positive effect on the macroscopic aspect of cutaneous lesions in cattle was observed during the first two treatment days [129].

In a recent paper, the authors developed membranes for wound healing dressing by combining a natural latex from *Hevea brasiliensis* (*Euphorbiaceae*) providing angiogenic action, with leaf ethanolic extract of *Casearia sylvestris Sw*. (diterpene concentrated fraction and casearin J) characterized by anti-inflammatory and wound-healing activities [130]. Permeation and retention assays demonstrated the dermal penetration of phenolic compounds from the membrane with ethanolic extract and penetration of casearin-like clerodane diterpenes from all membranes, indicating that these topical systems have great potential for therapeutical application of *C. sylvestris* components [130].

### 6.6. Crocus sativus

*Crocus sativus* L., from the family of *Iridaceae*, is commonly known as saffron, which is the name of a spice derived from the *C. sativus* stigmas dried. Saffron is used for medicinal purposes in Chinese, Ayurvedic, Persian, and Unani traditional medicines [131]. The therapeutic properties of saffron used for healing purposes could be found in Materia Medica, written by a Greek physician (*Pedanius Dioscorides*) in the first century A.D. [132]. Modern pharmacological studies have demonstrated that saffron extract or its bioactive constituents, such as apocarotenoids, mono-terpenoids, flavonoids, phenolic acids, and phytosterols, have a wide range of therapeutic effects [133]. Major carotenoids contained in saffron extracts (SE) include crocin, crocetin, picrocrocin, and safranal [134] (Figure 3). Crocin, the digentiobiosyl ester of crocetin, is one of the few water-soluble carotenoids found in nature [134]. These compounds have been shown to have a wide spectrum of biological activities, including several antioxidant and anti-inflammatory properties [133,134,135,136]. Recent studies have shown that the effects of crocin, crocetin, and safranal against oxidative stress include the reduction in lipid peroxidation (malondialdehyde levels) and nitric oxide levels, and an increase in the levels of glutathione, antioxidant enzymes (superoxide dismutase, catalase, and glutathione peroxidase), and thiol content [137,138,139,140].

Despite the large number of papers indicating that saffron and its carotenoids have potent antioxidant and anti-inflammatory properties, only a few studies have reported these effects in wound healing [141,142]. In an in vitro study, anti-inflammatory properties and ROS scavenging activity of crocin were investigated in human epidermal keratinocyte and human dermal fibroblasts, [143]. Crocin inhibited squalene peroxidation and UVA-induced cell membrane arachidonic acid peroxidation in a dose-dependent manner (31-113-195 µM) and, from 300 µM to 1 mM, prevented the release of inflammatory mediators, modulating the expression of NF-kB-related genes and glycosylation-related genes [143]. Given the impact of UV exposure on skin quality [144], this study suggests that crocin can be regarded as a molecule with potential beneficial effects against skin aging.

More recently, an in vivo study was conducted to compare the effect of saffron extract to silver sulfadiazine (SSD), the most widely used topical treatment for burn injuries, in a burn wound model in rats. The animals treated with 20% saffron pomade showed more significant wound closure than the SSD-treated and untreated groups (83.04 ± 1.36% versus 57.57 ± 2.8% versus 35.53 ± 3.5%, respectively, on day 7, *p* < 0.001), including re-epithelialization and wound contraction [145]. Based on the scratch test, in vitro experiments confirmed the positive role of saffron in fibroblast migration and proliferation during the remodeling phase, as well as regenerative and anti-scarring properties of saffron [145].

In the production of saffron spices, copious amounts of floral bio residues are generated and wasted. Interestingly, polyphenols and anthocyanins found in the sepals and stamens of *C. sativus* have been used as a source of antioxidant and anti-inflammatory molecules with beneficial effects against skin pathologies [146,147]. Khorasani and colleagues [142] previously evaluated the efficacy of saffron pollen extract in cream in the treatment of thermally induced burn wounds in rats. When compared to those treated with silver sulfadiazine, the results showed that saffron cream accelerated wound healing and reduced healing time. Furthermore, wounds treated with saffron cream had fewer inflammatory cells than the other groups, and histological studies of wound sections confirmed the formation of new epiderma, and fewer inflammatory cells infiltrated the dermis. These effects were most likely mediated by antioxidant and anti-inflammatory mechanisms, though the exact mechanism has yet to be determined [142].

Zeka et al. [141] developed a new hydrogel enriched with antioxidants derived from saffron flowers. When kaempferol or 10 µM crocin-enriched hydrogels were used in fibroblast cell cultures isolated from newborn mice (as a model for skin), it was found that cells grew faster when compared to the control [141]. Although more research is needed to understand the mechanisms of action and pathways involved in wound healing, the antioxidant potential of compounds extracted from saffron petals and other floral bio residues can be effectively used in wound management.

### 6.7. Curcuma longa

*Curcuma longa* is an herbaceous, perennial, rhizomatous plant belonging to the *Zingiberacee* family (one of the many species of the *Curcuma* genus). The spice is obtained from the rhizome of *Curcuma longa* (commonly called “turmeric”) and contains polyphenolic antioxidant compounds [148]. In Ayurvedic and folk medicine, turmeric has been used for a long time to treat various inflammatory diseases. The anti-inflammatory effect of turmeric is due to its ability to decrease the production of histamine and prolong the action of the natural anti-inflammatory adrenal hormone, cortisol [149].

Turmeric’s effects on health are generally centered upon its principal component, called “curcumin,” an orange-yellow-colored, lipophilic polyphenol compound [150] (Figure 3). Curcumin is one of the three curcuminoids present in turmeric, making up 2 to 5% of the spice [150] and approximately 77% of the singular extract [150]. The structure of curcumin (1,7-bis (4-hydroxy-3methoxyphenyl)-1,6-hepadiene-3,5-dione) was first described by Milobedska et al. in 1910 [151]. Curcumin is a highly pleiotropic molecule [152]. In recent times, it has been extensively studied as an anti-cancer [153], anti-aging [154], antiviral [155], antibacterial [156], and wound-healing agent [157]. The wound-healing potential of curcumin is attributed to its anti-inflammatory [158], antioxidant, and radical scavenging [159] activities. Curcumin acts in all wound-healing stages. In the inflammation stage, curcumin was shown to inhibit the activity of NF-kB and the production of TNF-alpha and IL-1 [160,161].

Radical-scavenging activities of curcumin have been reported in studies on wound healing, although this compound can act as a pro-oxidant when used at high doses [162,163,164,165]. In in vitro studies on keratinocytes and fibroblasts [166,167], curcumin provided optimal protection against hydrogen peroxide. In the proliferative stage, curcumin enhances fibroblast migration, granulation tissue formation, collagen deposition, and re-epithelialization. Curcumin improves wound contraction during the remodeling stage by increasing TGFbeta-production and, as a result, fibroblast proliferation. The suppression of acute and chronic inflammation also occurs by blocking the formation of cyclooxygenases, lipoxygenases, and inducible nitric oxide synthase enzymes [166]. 

Curcumin was used in a recent study to optimize the survival and tolerance capacity of bone marrow mesenchymal stem cells, which were used to improve tissue repair in chronic wounds and burns, in terms of self-renewal and paracrine factor production [167].

Furthermore, curcumin was explored for potent antiherpetic actions against herpes simplex virus type 1 and type 2, and the associated inflammation connected to these diseases [155]. Indeed, clinically, HSV-1 and HSV-2 infections cause lesions on the mucocutaneous junctions of the face and genitalia. Vesicular lesions can sometimes ulcerate, leaving recalcitrant wounds that are difficult to treat [155]. Until now, the foundation of treatment has been the eradication of viral infection. Little attention has been paid to the outcome of the viral infection and the wounds that resulted, specifically whether this represents an epidermal or dermal injury. Curcumin represents an opportunity for dressing and healing these lesions.

Unfortunately, the use of curcumin is limited by its low bioavailability, rapid metabolism, poor solubility, and light sensitivity [152]. A large number of recent papers [168,169,170,171,172,173,174,175,176,177,178] reported the incorporation of curcumin into nanocomposite materials such as nanofibers, nanoparticles, hydrogels, or novel combinations of these. Recently, numerous researchers have developed biocompatible and biodegradable composite nano-fibrous materials and hydrogel systems containing curcumin for wound healing applications, for example as an anti-scar or in chronic wounds [166,169]. These scaffold systems include polyvinyl pyrrolidone (PVP) [169], cerium nitrate hexahydrate (Ce(NO3)3·6H2O) [170], nanofibers of carboxymethyl guargum (CMGG) [171], electrospun amine-functionalized SBA-15 (Santa Barbara Amorphous) incorporated with PVA (Poly Vinyl Alcohol) [172], nanofibers of cellulose acetate (CA) integrated with graphene oxide, TiO2 [173], electrospun fibers comprising SBA-15 (Santa Barbara Amorphous), amine-functionalized SBA-15 polycaprolactone (PCL) [174], nanofibers of electrospun poly(ε-caprolactone) (PCL), Chitosan (CS) [175], thermo-sensitive sodium alginate hydrogel cross-linked with poly(N-isopropylacrylamide), and curcumin (Alg-g-pNIPAM) copolymers [176]. All these nanofibrous scaffolds exhibited high porosity and biocompatible characteristics that enhance the solubility of this hydrophobic drug, aiding its release in a controlled manner [176]. A novel curcumin-loaded sandwich-like nanofibrous membrane was prepared using sequential electrospinning. The nanofibrous membrane in the sub-layer consisting of gelatin, chitosan, and PCL provided efficient hemostatic activity and was effective in absorbing exudate and keeping the wound moist. The curcumin-loaded nanofibrous membrane in the mid-layer released curcumin, therefore reducing wound oxidative stress and inflammation. Finally, the top layer, consisting of Ag nanoparticles, silk fibroin, and PCL, released Ag nanoparticles, which acted against external bacteria [176]. In a further study, when the eggshell membrane, a highly proteinaceous thin layer present between the egg white and shell, was added to the bottom layer made up of PVA with curcumin nanoparticles, antioxidant and anti-inflammatory action was implemented with positive effects on the migration of dermal fibroblasts [177]. This patch contributed to wound healing by providing exudate absorption, releasing therapeutical components, and supporting the deposition of extracellular matrix [177,178].

### 6.8. Glycyrrhiza glabra L.

*Glycyrrhiza glabra* is a herbaceous perennial legume of the *Fabaceae* bean family, from whose root is extracted sweet and aromatic flavoring known as licorice. The licorice plant is widely used as an herbal remedy [179] and in skin-care products [180]. Numerous articles have reported pharmaceutical therapeutic properties, such as antifungal, antibacterial, antiviral, and antioxidant properties, but these are only some of the possible therapeutic properties [181]. Licorice root extract has been used for years as an effective medication, especially in gastric ulcers. The main biologically active components of licorice include terpenoids such as triterpene saponins, chalcones, and glycyrrhizin, flavonoids, and isoflavonoids, which are responsible for the observed activities [179] (Figure 3).

Some studies have demonstrated antiulcer effects of *G. glabra* in the healing of gastric [182], oral [183], and colitis mucosal ulcers [184], as well as corneal neovascularization [185], but only a few studies have reported its wound-healing activity on full-thickness dermal wounds [186,187,188,189,190]. Previous research has shown that antioxidant and anti-inflammatory compounds of *G. glabra*, such as triterpenes and flavonoids, reduce free radicals and pus in the wound area, aiding in wound healing [187,188].

Zangeneh et al. [189] discovered that 3% *G. glabra* aqueous extract ointment significantly accelerates wound-healing activity in rats by reducing the wound area, total cells, macrophage, lymphocyte, and neutrophil levels while increasing wound contracture, fibrocyte, hexuronic acid, and hydroxyproline levels when compared to the basal ointment and control groups.

According to the findings of Siriwattanasatorn et al. [190], ethanolic extract of *G. glabra* promoted proliferation and accelerated wound closure by inhibiting superoxide anion and nitric oxide production with IC_50_ values of 28.62 ± 1.91 and 46.35 ± 0.43 μg/mL, respectively. The bioactive compound, glycyrrhizin, exhibited antioxidant activities with an IC_50_ value of 40.85 ± 2.30 μg/mL but had no activity against nitric oxide production inhibition [190].

More recently, Hanafy et al. [191] studied the effect of *G. glabra* extract on full-thickness wound healing in a Guinea pig model. In this study, it was found that the topical application of a cream containing 5–10% *w*/*w G. glabra* extract accelerated wound healing with statistically significant differences in the re-epithelialization of treated wounds, which were likely due to the presence of flavonoids [191].

### 6.9. Malva sylvestris

The use of *Malva sylvestris*, a species of the mallow genus *Malva* in the family, has been documented since long ago. *M. sylvestris* is recommended for acne and skincare, as an antiseptic and emollient [192,193,194], and as an antimicrobial and anti-inflammatory agent for burn and cut wound healing [195,196,197]. The healing capabilities of this plant relate to the mucilage and flavonoids found in the leaves and flowers [198]. Indeed, *M. sylvestris* flowers extract contains anthocyanin, malvidin, flavones, flavonols, malvin, malvaline, niacin, and folic acid, which are responsible for their pharmacological and biological activities [197,198,199] (Figure 3).

Afshar et al. [200] investigated the wound-healing ability of *M. sylvestris* using a wound mouse model. The researchers reported that mice treated with silver sulfadiazine were less capable of wound healing than mice treated with 1% *M. sylvestris* extract, where increased collagen synthesis was observed [200]. Another in vitro study investigated the effect of *M. sylvestris* or silver sulfadiazine on wound healing using a second-degree burn model [201]. The results clearly demonstrated that both 5% and 10% *M. sylvestris* cream was superior to sulfadiazine in terms of reducing the time required for complete wound healing [201]. Nevertheless, the molecular mechanism underlying these actions requires further investigation.

Recent research has examined the effect of *M. sylvestris* extract inserted into a novel polyurethane (PU)-based nanofibers used as dressing for diabetic wounds [202]. When animal wounds treated with *M. sylvestris* nanofibers were compared to control groups, the wound area was shown to be significantly decreased. A polymer blend containing 15% *w*/*w* herbal extract of *M. sylvestris* was used to monitor the herbal compound’s progressive release over an 85-h period. In comparison to the control group, treatments with extract-loaded wound dressings were significantly more effective at reducing acute and chronic inflammation, increasing collagen deposition and neovascularization, and demonstrated acceptable antibacterial activity against *Staphylococcus aureus* and *Escherichia coli* [202].

### 6.10. Plantago

*Plantago L*. (*Plantaginaceae*) is a worldwide genus including approximately 260 species of annual and perennial herbs and shrubs [203]. Phytochemical studies have shown that the *Plantago* genus contains a great number of compounds such as acteosides, iridoids, glucosides, phenylethanoid glycosides, flavonoids, tannins, triterpenes, saponins, sterols, and phenyl carboxylic acid derivatives [204] (Figure 3). The genus *Plantago* is widely used in folk medicine as an anti-inflammatory agent for various diseases, including the treatment of wounds [205]. Fresh plantago leaves were shown to have antibacterial effects and to be beneficial for wound healing. Crushed leaves are used to treat chronic wounds, abscesses, and acne [206]. Not many studies have investigated the effects of *Plantago* species on wound healing.

An in vitro investigation of human oral epithelial cell lines H400 indicated the anti-inflammatory properties of *P. major* are due to the synergistic actions of polyphenols and water-soluble substances (i.e., polysaccharides) [206]. Genc et al. investigated the wound healing ability of *P. subulata* in vitro using fibroblasts (L929 cell line), whereas the anti-inflammatory activity was evaluated using macrophages (RAW 264.7 cell line) [207]. Treatments with different concentrations (200, 400, 800, and 1000 μg/mL) of the aqueous fraction of *P. subulata* methanolic extract (PHE) exerted protection against oxidative stress in fibroblasts, although not in correlation with PHE concentrations. Preincubation of macrophages with 50 and 200 μg/mL PHE reduced the production of NO, PGE2, and TNF-α induced by lipopolysaccharide [207]. Acteoside was thought to be the chemical responsible for these effects [207].

Recently, a clinical investigation assessed the efficacy of a gel comprising *A. vera* and *P. major* in treating diabetic foot ulcers [208]. At the conclusion of the randomized open-label controlled trial, patients treated with a topical gel containing 10% plantago hydroalcoholic extract applied to the wound once daily for two weeks showed a significant reduction in wound size when compared to the control group [208]. The application of *P. major* topical gel accelerates the healing of diabetic foot ulcers and pressure ulcers by reducing the wound’s erythema. Additionally, the proportion of patients who completed the wound-healing process was higher in those who received *P. major* dressing [208]. However, this study was limited by its relatively small sample size, absence of placebo and blinding (due to technical issues), short duration of intervention, and use of novel dressings in the control groups (because of different characteristics of ulcers necessitating this). These results showed that Plantago and its compounds may be good candidates for future drug studies.

### 6.11. Salvia officinalis

*Salvia officinalis**L.* (also called **sage,** garden sage, or common sage) is a perennial evergreen subshrub with woody stems, grayish leaves, and blue to purplish flowers [209,210]. *Salvia* is an important genus consisting of approximately 900 species in the family *Lamiaceae* (formerly *Labiatae*). *Salvia* species are native to the Mediterranean region and are cultivated worldwide for use in folk medicine and culinary purposes. The name Salvia is derived from “*Salvere*”, a Latin word meaning “to feel well and healthy” [211]. The genus *Salvia* has a wide range of medicinal uses, and it is traditionally used to treat more than sixty health illnesses [211,212].

According to the analysis of essential oils taken from aerial sections of the plant, these species contain approximately 120 phytocomponents [212,213,214,215,216], including fatty acids, carbohydrates, alkaloids, glycoside saponins, terpenes, phenolic acids, flavonoids, and polyacetylenes [212,215]. The biological actions of oil and aqueous extracts include anti-inflammatory, antibacterial, and antioxidant activity [216,217,218]. The terpenoids, e.g., 1,8-cineole, oxygenated sesquiterpene viridiflorol, camphor, nonacosane, and pentacosane, were identified as the major constituents of the essential oil [219] (Figure 3). Numerous studies have demonstrated that the composition of sage essential oil undergoes considerable changes in relation to seasonal variation, genetic diversity, defense in plant sections, and developmental stage differences [218,220].

Numerous investigations have established the influence of *S. officinalis* on wound healing [220,221,222,223,224]. Farahpour et al. [222] demonstrated that 2% and 4% (*w*/*w*) topical ointments of sage oil on mouse full-thickness cutaneous wounds dramatically decreased pro-inflammatory cytokines and the overall bacteria count when compared to a control group [222]. The data in this study indicated that this treatment suppressed the pro-inflammatory response by decreasing mRNA expression levels of IL-6, IL-1, and TNF-alpha, and promoted fibroblast proliferation by increasing cyclin-D1 expression [222]. Furthermore, *S. officinalis* has been shown to stimulate VEGF transcription and secretion and FGF-2 expression at the wound edge and in basal keratinocytes [222]. In in vivo experiments using a wound model in rats, topical application of 1, 3, and 5% *S. officinalis* hydroethanolic leaf extracts significantly increased, particularly at higher doses, the percentage of wound contraction, the re-epithelialization period, the breaking strength ratio, and upregulated hydroxyproline content compared to the control group [222]. Additionally, *S. officinalis* significantly increased new vessel formation and fibroblast distribution [223].

Another study has looked at the effects of ethanol leaf extracts of two different *Salvia* species, *Salvia kronenburgii* (SK) and *Salvia euphratica* (SE), at two different concentrations (0.5% and 1% (*w/w*) on incision and excision wound models in diabetic rats [224]. In addition, these extracts were tested for their ability to fight off bacteria, viruses, and fungi. They worked best against *Staphylococcus aureus* and *Bacillus subtilis*, as well as *Escherichia coli*, *Aeromonas hydrophila*, and *Mycobacterium tuberculosis*. [224]. Moreover, both concentrations of SK and SE ointments were found to be as effective as the reference drug, showing significant wound closure ratios with a range of 96.9–99.9% in excision and incision wound models after the 14th day [224]. These results suggest that Salvia species, which are common in nature, may be promising antimicrobial and wound-healing agents for the treatment of infectious diseases as an alternative to synthetic drugs with high costs and side effects.

### 6.12. Rosmarinus officinalis (Syn. Salvia rosmarinus)

*Rosmarinus officinalis* L., commonly known as rosemary, is a member of the *Lamiaceae* family. Recent evolutionary research has reported that the genus *Rosmarinus* has merged with the genus *Salvia*. On this basis, *Rosmarinus officinalis* was changed into *Salvia Rosmarinus* [225]. Rosemary is a fragrant, needle-like-leaved plant that is widely cultivated worldwide and used in folk medicine. Due to the presence of carnosol/carnosic and ursolic acids, rosemary has therapeutic properties and is used in the pharmaceutical and cosmetics industries, primarily for its antioxidant and anti-inflammatory properties [226]. Other uses, including treatments for wound healing, skin cancer, and mycoses, have also been investigated [227,228,229]. Potential applications in the treatment of non-pathological skin conditions, such as ultraviolet damage and aging, have been shown [228,229,230,231].

Rosemary contains a large number of secondary metabolites that have been identified using ultra- and high-performance liquid chromatography and gas chromatography to reveal significant amounts of phenolic compounds (diterpenoids and flavonoids) and volatile chemicals [232,233]. The flavonoids discovered in rosemary (eriocitrin, luteolin 3′-O-D-glucuronide, hesperidin, diosmin, isoscutellarein 7-O-glucoside, hispidulin 7-O-glucoside, and genkwanin) were found in various parts of the plant over time [233] (Figure 3).

In an in vivo study, the group treated with rosemary essential oils demonstrated significant improvements in healing, angiogenesis, and granulation tissue formation when compared to the control group [230]. Another study reported that diabetic and non-diabetic rats treated topically with 10% *R. officinalis* oil experienced rapid wound healing [234]. Using a rat excision wound model, a recent in vivo study examined the wound healing potential of three chitosan-based topical formulations containing either tea tree essential oil, rosemary essential oil, or a combination of the two oils [235]. When compared to groups treated with individual essential oils or the control group, topical application of a chitosan-based product containing a blend of both oils increased wound shrinkage significantly. Additionally, histopathological examination revealed complete re-epithelialization and activation of hair follicles when the two essential oils were combined. Monoterpene content of essential oils contributed significantly to their antioxidant and wound-healing properties [235]. In conclusion, rosemary extract has a great deal of therapeutic potential and could help wounds heal at different stages.

## 7. Conclusions and Future Direction

Wounds are severe conditions that negatively impact the quality of life of people worldwide. Effective treatment of wounds depends upon the interaction of appropriate cell types, cell surface receptors, and the extracellular matrix with the therapeutic agents. Due to the complexity of skin tissue structure, the development of an ideal medication, which can lead to a rapid and effective healing process, remains a challenge.

Scientific evidence obtained in the last five years has allowed us to expand our knowledge about the effects of herbal medicaments on wound healing and underlying molecular mechanisms. All the plants reported in this review exert antioxidant, anti-inflammatory, and antimicrobial activity. Interestingly, antiviral activity has been detected in *B. striata* and *C. longa* (see Table 1). Recent literature has provided new evidence that the activity of herbal medicaments active in wound healing is mainly ascribed to flavonoids, alkaloids, saponins, phenolic compounds, and polysaccharides, which can act at different stages of the process by means of various mechanisms. Thanks to increased investigations of mechanistic events, polyphenolic compounds have been confirmed as therapeutical agents in wound healing by controlling and modulating inflammatory responses. Many phytochemicals, such as triterpenoids, curcumin, etc., present in medicinal plants were found to be key factors in homeostasis, re-epithelialization, and regeneration by promoting fibroblasts proliferation and/or collagen production. Carotenoids, flavonoids, and triterpenoids in wound dressings inhibit oxidative stress and promote antioxidant activity. Reactive oxygen species (ROS), by acting as secondary messengers to immunocytes and non-lymphoid cells, are involved in the repair process, the recruitment of lymphoid cells, angiogenesis, and the destruction of pathogens at the wound site. Elevated ROS have been detected in vivo and have been associated with impaired wound repair [167]. In addition to promoting pro-inflammatory cytokine secretion and the induction of matrix metalloproteases, excessive ROS can modify and/or degrade extracellular matrix proteins with negative effects on dermal fibroblasts and keratinocytes [169].

There has been an increasing interest in the synergistic effect of different extracts with specific phytocomponents. For example, when *B. striata*’s extract polysaccharides, primarily glucomannans, are enriched with *B. striata* extract mainly containing polyphenols, more effective healing of wounds is observed [106].

From this review of the literature, a lack of uniformity in the experimental design and protocols clearly emerges, including the methods of extraction, administration modality, and experimental models (Table 1). Therefore, important issues in the use of phytochemicals in wound healing are the definition of optimal doses and the route of administration. To this end, research should focus on establishing the most effective phytochemicals mixture, the best plant source, and the mode of administration.

Current trends in wound care are moving toward the development of innovative wound care treatments that combine the use of herbal healing agents with modern products and practices. In recent years, nanostructures and nanoformulations have promisingly overcome the drawbacks of common medicaments. They regulate the release of therapeutics, reduce the doses required for healing, and improve the solubility and wound-healing efficiency of water-insoluble herbal components. The well-regulated porosity and the similarity of nanofibers to the skin tissue make them the ideal dressing for wound management. Researchers have investigated the incorporation of natural substances into nanofibrous architectures for wound dressing, such as composite sponge made of *B. striata* polysaccharides and its extract, alginate hydrogels loaded with *A. vera*, dressing films with *C. officinalis* extracts, hydrogel sheets containing saffron components, as well as nanofibrous scaffold containing curcumin. The possibility of using a biocompatible formulation made of natural lipids or a polysaccharide matrix and herbal extracts would give the consumer a “green” option and almost no side effects once it was put on the skin.

Based on these observations, different therapeutic strategies should be used simultaneously in the management of wounds, especially chronic wounds, to accelerate the wound-healing process and avoid wound complications. Furthermore, to increase the efficacy and use of natural substances in wound care some issues remain to be solved. Multidisciplinary efforts are required to establish the products’ safety, investigate their side effects, and conduct double-blind controlled clinical trials. Good manufacturing practices and regulatory legislation also play a critical role in increasing clinicians’ use of phytotherapy and promoting its integration into national health systems.

## Figures and Tables

**Figure 1 molecules-27-03566-f001:**
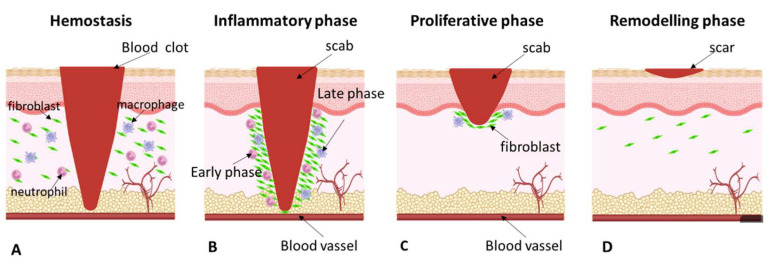
Wound-healing process. The wound healing process is commonly categorized into four distinct phases: (**A**) Homeostasis (coagulation); (**B**) inflammatory phase (early/ late inflammation); (**C**) proliferative phase (proliferation/migration/epithelialization/granulation); (**D**) remodeling phase (maturation/repair). (**A**) Hemostasis: A clot is formed, providing a temporary barrier to fluid loss and pathogen entry; acts as a reservoir of bioactive factors and antimicrobials; provides provisional extracellular matrix, which supports immune cell infiltration and migration; and initiates tissue repair pathways. (**B**) Inflammatory phase: Early step with damage-associated molecular patterns activation, free radicals, and reactive molecular species production to recruit immune cells; release of antimicrobial species; infiltrating immune cells that secrete amplifying alarmin signals (endogenous, constitutively expressed, chemotactic, and immune-activating proteins/peptides that are released as a result of degranulation, cell injury or death, or in response to immune induction), and activation of keratinocytes and fibroblasts. (**C**) Proliferation phase: Migration and proliferation of keratinocytes, fibroblasts, endothelial; resolution of inflammation; collagen/extracellular matrix synthesis; decreased vessel permeability; new capillary and lymphatic vessel angiogenesis; epithelialization; and de novo formation of granulation tissue. (**D**) Remodeling (maturation): Collagen/extracellular matrix turnover (synthesis and degradation); extracellular matrix reorganization and realignment; extracellular matrix contraction; endothelia and fibroblast apoptosis; repigmentation.

**Figure 2 molecules-27-03566-f002:**
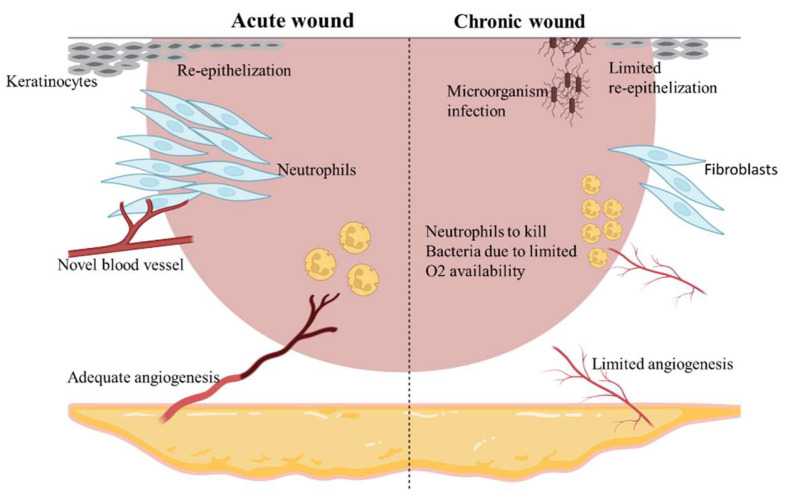
Processes active in acute and chronic wounds. Acute wounds (left side): Adequate angiogenesis with re-epithelialization promotion, fibroblasts’ proliferation, and neutrophils’ anti-infection activities. Chronic wounds (right side): Persistent local bacterial infections, failure formation of novel blood vessels. decreased fibroblasts’ proliferation, and the neutrophils’ anti-infection activities hampered by poor angiogenesis.

**Figure 3 molecules-27-03566-f003:**
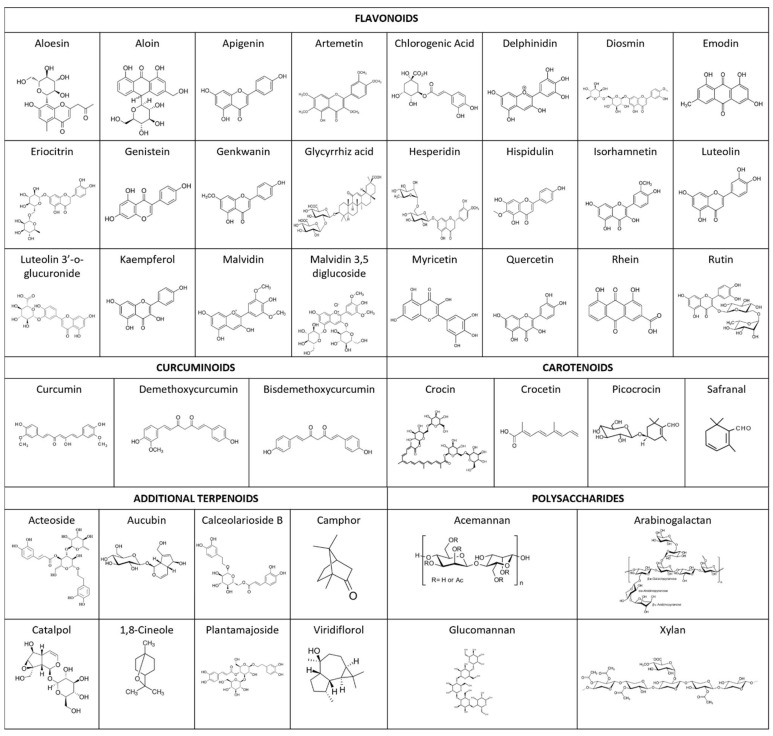
The primary bioactive phytochemical structures involved in wound healing.

**Table 1 molecules-27-03566-t001:** Medical plants and their effects in wound healing.

Plant	Parts Used	Main Bioactive Components	In Vivo Wound Model, Doses and Routes of Administration	In Vitro Model and Doses	Biological Activity	Mechanism of Action	Ref.
*Achillea millefolium*	flower aerial parts	FLAVONOIDS (chlorogenic acid, apigenin, artemetin, luteolin, quercetina and shaftoside)	Full-thickness incisional wound in Sprague-Dawley rats―topical: 3% acqueous extract (AAE).	Hs68, HaCaT, and RAW 264.7 cell line: 25–100 μg/mL AAE.	Antibacterial Anti-inflammatory Re-epithelialization process	Modulates the inflammatory cytokine and growth factor; activates Akt signaling pathways; stimulates collagen expression; stimulates keratinocyte differentiation and motility; reduces inflammatory mediators NO and PGE_2_.	[69]
*Aloe vera*	fleshy leaves	FLAVONOIDS (aloin, aleosin, emodin, rhein) POLYSACCHARIDES (acemannan, acetylated polymannan and glucomannan)	Full-thickness wound in Wistar rats― topical: 25–50 mg/mL in gel. Full-thickness wounds in hairless mouse― topical: 0.1% and 0.5% *w*/*w*. Full-thickness wounds in mice― topical: 10 and 50 mg/kg. Burn wounds in BALB/c mice― Topical: aloe-emodin 1 × 10^−8^, 1 × 10^−10^ and 1 × 10^−12^% *w*/*w*.	Fibroblast: 50, 100 and 150 μg/mL. HaCaT and Raw 264.7 cell line: 1, 5 and 10 µM aloesin. Primary epidermal keratinocytes and dermal fibroblasts: 2% and 3%	Antibacterial Anti- Inflammatory Re-epithelialization process	Modulates the inflammatory response; modulates signaling proteins phosphorylation; Stimulates collagen deposition and angiogenesis; Strongly promotes fibroblast proliferation and moderately stimulates keratinocyte migration.	[77,78,79,81,83,87]
*Bletilla striata*	root (tuber)	FLAVONOIDS (anthocyanins) POLYSACCHARIDES (glucomannan) TRITERPENOIDS STILBENOIDS (bibenzyl, bletilol D, bletilol E, dihydrophenanthrene and phenanthrene)	Partial-thickness burn wound model in mice topical: 1 mg/mL BSP extract or BSP polysaccharides residue extract or Mix	human mesangial cells (HMCs): 5, 10, and 20 μg/mL.	Antimicrobial and antiviral Antioxidative Anti-aging Anti-inflammatory Re-epithelialization process Hemostatic activity	Promotes expression of mediators of the inflammatory response (TNF-α, IL-1β, and IFN-γ); increases NO and promotes neutrophils, monocytes, and macrophages chemotaxis; promotes epithelial cells growth and fibroblast proliferation.	[91,97,101,103]
*Calendula officinalis*	flowers	TRITERPENOIDS FLAVONOIDS (rosmarinic acid, caffeic acid, 5-*O*-caffeoylquinic acid, isorhamnetin-3-*O*-glucoside, isorhamnetin-3-*O*-rutinoside, kaempferol-3-*O*-rutinoside, quercetin-3-*O*-glucoside and quercetin-3-*O*-rutinoside) COUMARINES, QUINONES	Full-thickness excisional wound in BALB/c―topical: 150 mg/kg BW ethanolic or water extract ointment. Metallic punch Wistar rats—topical: 100 uL of acqueous solution of 1% ethanolic extract. Incisional wound in Sprague-Dawley rats― topical: 5–10% gel. Full-thickness wound in Wistar rats topical: wound dressing in nanofibers with 2% CO extract.	Human dermal fibroblasts: 100 ug/mL of extract. NIH-3T3, WI38, human dermal fibroblasts: 1:100–1:50 tinture for 0–60 min. human immortalised Keratinocytes: 10–50 ug/mL of hexanic, ethanolic or acqueous extracts.	Anti- Inflammatory Re-epithelialization process	Promotes expression of mediators of the inflammatory response; increases keratinocytes and fibroblasts proliferation; stimulates collagen production and angiogenesis; inhibits lipoxygenase activity; reduced glutathione levels.	[110,111,115,116,117,118,119]
*Casearia sylvestris*	leaves	TRITERPENOIDS (clerodane diterpenes) PHENOLIC ACIDS	Full-thickness lesions― topical: 0.1, 0.3, 1.0 mg/site extract. Second degree burn in Wistar rats―topical: biofilm with 1 g of lyophilized extract or spray with extract.		Anti-inflammatory	Reduces early and late edema; reduces myeloperoxidase activity.	[123,127,128]
*Crocus sativus*	stigmi, stamen, flowers	CAROTENOIDS (crocin, crocetin, picrocrocin and safranal) MONO-TERPENOIDS FLAVONOIDS (kempherol and quercetin) PHENOLIC ACIDS	Second degree burn in Wistar rats―topical: cream with 20% pollen saffron. Full thickness wound in Sprague-Dawley rats―topical: pomade with 20% saffron extract	Fibroblasts from newborn mice: hydrogel with 160 mg/L crocin from saffron. Human dermal fibroblasts: 3.12–50 μg/mL for 6–24 h C2C12, MCF7, HCT116 cell lines:125 ug/mL of saffron anther extract	Antioxidant Anti-inflammatory Re-epithelialization process	Reduces the level of pro-inflammatory cytokines (TNF-α and IL-6) Increases level of anti-inflammatory cytokines (IL-4 and IL-10); Inhibits lipid peroxidation Enhances vascularization; increases fibroblasts proliferation.	[141,142,144,145]
*Curcuma longa*	rhizomes	CURCUMINOIDS (bisdemethoxycurcumin, curcumin and demethoxycurcumin)	Full-thickness wound in Balb/c mice―topical: gel 3% curcumin Full-thickness wound in W rats―topical: PCL nanofibers 10% curcumin Full-thickness wound in W. Rats―topical: PVA nanofibers 1% curcumin Full thickness wound model in SD rats―topical: 100–200 ug/mL curcumin nanoparticle loaded dermal patch	Rat BMSCs bone marrow mesenchymal stem cells: 10 μM curcumin Mouse 3T3 fibroblast: PCL nanofibers 10% curcumin Swiss 3T6 cell lines PVA nanofibers 1% curcumin HADF cells: 100–200 ug/mL curcumin nanoparticles	Antioxidant Radical-Scavenging Anti-inflammatory Re-epithelialization process	Regulates many genes implicated in the initiation of inflammatory responses (NF-(κ)B, AKT, PI3K, IKK); enhances fibroblast migration, granulation tissue formation, collagen deposition; increases TGF-β production; increases fibroblast proliferation.	[167,169,171,175,177]
*Glycyrrhiza glabra*	root, leaves	FLAVONOIDS TERPENOIDS (glycyrrhizic acid, saponins and triterpene) CHALCONES (glycyglabrone & licochalcone C)	Sprague Dawley rats wound―topical: 3% extract in crem. Guinea pig full-thickness wound―topical: 5% and 10% extract in cream.	3T3 Cell Line: 1–25 μg/mL extract	Antimicrobial Anti-inflammatory Antioxidant Re-epithelialization process	Increases collagen deposition; increases the wound healing rate; reduces superoxide anion; inhibits NO production; increases fibroblast proliferation.	[189,190,191]
*Malva sylvestris*	flowers, leaves, root, whole plant	POLYSACCHARIDES FLAVONOIDS (malvidin, malvin, delphinidin, genistein; myricetin, apigenin, quercetin and kaempferol) TERPENOIDS (monoterpenes, diterpenes, sesquiterpenes and nor-terpenes)	BALB/c mice cut wound– topical: 1% extract in cream. Second degree burn wound in rats―topical: 1–5–10% extract in cream. Diabetic streptozotocin induced wound in W rats—topical: 5–20% extract containing nanofibers	Mesenchimal Stem cells: 5–20% extract containing nanofibers	Antibacterial Antioxidant Anti-ageing Anti-inflammatory	Modulates the inflammatory response; Increases collagen deposition; Enhances vascularization; increases the wound healing rate.	[200,201,202]
*Plantago* L.	leaves	MONOTERPENOIDS (aucubin, acteoside, calceorioside B, catalpol, homoplantaginin and plantamajoside)		Oral epithelial cell line H400: 0.1 mg/mL extract L229 fibroblast cell line: 0.2–1 ug/mL extract Macrophages: 50–200 ug/mL	Antibacterial Antioxidant Anti-inflammatory	Inhibits NO production; reduces superoxide anion; reduces pro-inflammatory cytochine level (PGE_2_, TNF-α); decreases fibroblasts H_2_O_2_ cytotoxicity	[206,207]
*Salvia officinalis*	aerial parts	TERPENES (1,8-cineole) OXY-SESQUITERPENES (camphor, nonacosane and pentacosane viridiflorol)	BALB/c mice excisional splinting model―topical: 0.5% *w*/*w* dry extract in cream. BALB/c mice full-thickness wounds―topical: 2% and 4% essential oil ointment Wistar rats wound models: topical: 1%, 3% and 5%. hydroalcoholic extract Excision on Streptozotocin-induced diabetic rats: topical: 0.5% and 1% essential oil	Human dermal fibroblasts and epidermal keratinocytes: 0.031% extract	Anti-inflammatory Antimicrobial Antioxidant	Reduces pro-inflammatory cytokines; downregulates mRNA expression levels of IL-6, IL-1 β and TNF-α augments fibroblast proliferation via enhancing cyclin-D1 expression.	[216,221,222,223,224]
*Rosmarinus officinalis*	leaves, flowers roots, stems	FLAVONOIDS (diosmin, eriocitrin, genkwanin isoscutellarein 7-*O*-glucoside, hispidulin 7-*O*-glucosidehesperidin and luteolin 3-*O*-β-D-glucuronide)	full-thickness excision cutaneous wounds in alloxan-induced-diabetic BALB/c mice―topical: 100% essential oil - intraperitoneal injection: 0.2 mL, 10% (*v*/*v*) Excision on Streptozotocin-induced diabetic rats: topical: 100% essential oil. full thickness excision wound in Sprague Dawley rats -topical: 10% rosemary essential oil in chitosan	RAW 264.7 murine macrophage cells: 5–10 μg/mL hydroalcoholic extract	Antimicrobial Antioxidant Anti-inflaflammatory	Inhibits NO production; reduces inflammatory cytokines expression (IL-1β, IL-6, TNF-α); reduces expression of iNOS, COX-2, P-IκB and NF-κB/p65.	[228,230,232,234,235]

Akt: Protein Kinase B; NO: Nitric Oxide; TNF-α: Tumor necrosis factor alpha; IL: Interleukin; IFN-γ: Interferon gamma; NF-(κ)B: Nuclear Factor Kappa B; PI3K: Phosphoinositide 3-kinase; IKK: IkappaB Kinase; TGF-β: Transforming Growth Factor beta; PGE2: Prostaglandin E; H_2_O_2_: Hydrogen peroxide; iNOS: Inducible Nitric Oxide Synthase; COX-2: Cyclooxygenase-2; P-IκB: Phosphorylated Inhibitor kappaB.

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
