# Peer review of "Phytochemistry and Biological Activity of Medicinal Plants in Wound Healing: An Overview of Current Research"

_molecules, 2022, doi:10.3390/molecules27113566_

Round 1

Reviewer 1 Report

However, the manuscript presents a good topic, it needs major improvement before further consideration. Therefore, I recommend that the authors' team consider the following points to improve the quality of the manuscript.

  1. In the introduction section, please provide the keywords that have been used during the literature search to collect the reviewed studies. This is to ensure that the authors have collected all relevant data. 
  2. In Table 1, please replace the pictures of the presented plants with real ones or delete them. Using real images of plants in review papers represents a professional scientific style. 
  3. Please replace figures 3 and 4 with figures with higher resolution. Also, I recommend the authors re-draw all chemical structures in figures 3-7 to keep them in one style. It looks as if they were copied-pasted from some other sources. 
  4. Section (5.7 Curcuma longa), lines 631-633. I recommend the authors add a reference (DOI: 10.3390/microorganisms9020292), where curcumin has been reported to induce protective and therapeutic effects against herpesviruses and their associated inflammation. Herpes simplex virus is known to cause skin infection and skin ulcers. 
  5.  Most importantly, I recommend that the authors analyze and discuss the collected data critically. In other words, the reviewed studies and the acquired data should be deeply discussed and interpreted. Unfortunately, the manuscript in its current form looks like a hasty report, where, the manuscript just reports findings without assessing whether the studies are valid. Moreover, the authors should try to find (in the reviewed studies) all concentrations or doses that are accountable for the induced wound healing activities. These points should be carefully considered by the authors. 
  6. It would be better to add a new section that discusses the future direction and which strategies (biotechnological, genetic, etc.,) should be employed for designing effective and safe herbal products for treating wounds. 
  7. Please italicize all scientific names and check the whole manuscript, where all scientific names (family, genus, species, and variety or subspecies) should be italicized. 
  8. Finally, I recommend the authors double-check the whole manuscript for grammatical and typing errors and seek the help of a native English speaker to edit the English usage.

Author Response

Reply to reviewer 1

[GENERAL COMMENT] However, the manuscript presents a good topic, it needs major improvement before further consideration. Therefore, I recommend that the authors' team consider the following points to improve the quality of the manuscript.

RESPONSE: Thank you very much for your precious comments that helped us improve this manuscript.

[COMMENT 1.] In the introduction section, please provide the keywords that have been used during the literature search to collect the reviewed studies. This is to ensure that the authors have collected all relevant data. 

RESPONSE: We addressed this criticism by including a new section entitled "Methods" following the "Introduction," in which we detailed all methods used for literature searches and the associated keywords used during the literature collections. [Pg 2; Ln 70-84]

[COMMENT 2.] In Table 1, please replace the pictures of the presented plants with real ones or delete them. Using real images of plants in review papers represents a professional scientific style. 

RESPONSE: In table 1, the photos of the plants have been deleted since we lacked the time necessary to provide authentic, high-quality images of plants. [Pg 6]

[COMMENT 3.] Please replace figures 3 and 4 with figures with higher resolution. Also, I recommend the authors re-draw all chemical structures in figures 3-7 to keep them in one style. It looks as if they were copied-pasted from some other sources. 

RESPONSE: Figures 3,4,5,6,7,8 have been replaced with a new figure containing redrawn chemical 2D structures of principal bioactive compounds cited in the review. [Pg 8; Ln 266-268]

[COMMENT 4.] Section (5.7 Curcuma longa), lines 631-633. I recommend the authors add a reference (DOI: 10.3390/microorganisms9020292), where curcumin has been reported to induce protective and therapeutic effects against herpesviruses and their associated inflammation. Herpes simplex virus is known to cause skin infection and skin ulcers. 

RESPONSE: We added the reference suggested with relative comments [Pg 16; Ln 648; 684-692).

[COMMENT 5.] Most importantly, I recommend that the authors analyze and discuss the collected data critically. In other words, the reviewed studies and the acquired data should be deeply discussed and interpreted. Unfortunately, the manuscript in its current form looks like a hasty report, where the manuscript just reports findings without assessing whether the studies are valid. Moreover, the authors should try to find (in the reviewed studies) all concentrations or doses that are accountable for the induced wound healing activities. These points should be carefully considered by the authors. 

RESPONSE: We have taken into account this point by drawing a specific conclusion regarding the data reporting in each section and adding concentrations or doses where it was possible. [Ln 261; 317; 342; 388; 424; 433; 464; 537-539; 577-580; 591, 619; 750, 763; 772-774; 791; 802; 825; 831; 865; 875; 88; 917]

[COMMENT 6.] It would be better to add a new section that discusses the future direction and which strategies (biotechnological, genetic, etc.,) should be employed for designing effective and safe herbal products for treating wounds. 

RESPONSE: The Conclusion section has been revised by considering additional points and future directions, in a new section entitled “Conclusions and future directions”. [Pg 21; Ln 929-973]

[COMMENT 7.] Please italicize all scientific names and check the whole manuscript, where all scientific names (family, genus, species, and variety or subspecies) should be italicized. 

RESPONSE: We have checked and replaced with the italic character all the plants scientific names.

[COMMENT 8.] Finally, I recommend the authors double-check the whole manuscript for grammatical and typing errors and seek the help of a native English speaker to edit the English usage.

RESPONSE: We went through the entire manuscript to eliminate grammatical and typing mistakes.

Reviewer 2 Report

This review deals with the wound healing activity of a number of medicinal plants. The authors compiled information on the wound healing process, and listed 12 selected medicinal plants with previous in vitro and in vivo wound healing activity.  

Are these the only plants with reported wound healing activity? Please justify how the selection was done and only the papers in the last 5 years were used.

A lot of information has been compiled but there is a lack of thorough analysis and interpretation of the data. Moreover, there is excessive introductory information on the background and description of each plant but the actual data pertaining to the wound healing activities are very limited. 

The content of this review doesn't match its title (the word "current research") - there no emphasis on what's the current status of knowledge regarding the wound healing of these plants (and medicinal plants in general). 

Data presentation could be improved. Please standardize the format and quality of the chemical structures.

Permission must be obtained for the use of the photos of the plants in the table.

This manuscript, in its current form, is not comprehensive and does not advance our understanding on the topic.

Author Response

Reply to reviewer 2

[COMMENT 1] This review deals with the wound healing activity of a number of medicinal plants. The authors compiled information on the wound healing process and listed 12 selected medicinal plants with previous in vitro and in vivo wound healing activity.  Are these the only plants with reported wound healing activity? Please justify how the selection was done and only the papers in the last 5 years were used.

RESPONSE: Thank you for your insightful comments. While we are aware that there is an enormous variety of plant species having wound healing properties, (about 450 plant species have been identified), as we report in the new “Methods” section, we have chosen those most used in traditional medicine and of which there were found recent publications not yet reported in a review, so, we have decided to review the newest literature by choosing the publications of the last 5 years.

[COMMENT 2] A lot of information has been compiled but there is a lack of thorough analysis and interpretation of the data. Moreover, there is excessive introductory information on the background and description of each plant but the actual data pertaining to the wound healing activities are very limited. 

RESPONSE: The actual data pertaining to the wound healing activities are referred only to results published in the last five years, considering that the choice of plants was made in advance on the basis of the efficacy ascertained in wound healing.

[COMMENT 3] The content of this review doesn't match its title (the word "current research") - there no emphasis on what's the current status of knowledge regarding the wound healing of these plants (and medicinal plants in general). 

RESPONSE: We think that "current research" refers to the most recent and up-to-date publications in the literature due to this topic has been revised many times during the last 20 years.

[COMMENT 4] Data presentation could be improved. Please standardize the format and quality of the chemical structures.

RESPONSE: Figures 3,4,5,6,7,8 have been replaced with a new figure containing redrawn chemical 2D structures of principal bioactive compounds cited in the review.

[COMMENT 5] Permission must be obtained for the use of the photos of the plants in the table.

RESPONSE: In table 1, the photos of the plants have been deleted since we lacked the time necessary to provide authentic, high-quality images of plants.

[COMMENT 6] This manuscript, in its current form, is not comprehensive and does not advance our understanding on the topic.

RESPONSE: We addressed this criticism by thoroughly revising the test and correcting errors, drawing a specific conclusion regarding the data reporting in each section. The conclusion section has been changed to include more points and future directions in a new section called "Conclusions and Future Directions." [Pg 22]

Reviewer 3 Report

Interesting review. I wonder why authors have choosen only several species among hundreds  with this kind of activity. nevertheless is interesting review.

Author Response

Reply to Reviewer 3

[COMMENT] Interesting review. I wonder why authors have choosen only several species among hundreds with this kind of activity, nevertheless is interesting review.

RESPONSE: Thank you for your comments. While we are aware that there is an enormous variety of plant species having wound healing properties (about 450 plant species have been identified), we have chosen those most used in traditional medicine and of which there were found recent publications not yet reported in a review, so, we have decided to review the newest literature by choosing the publications of the last 5 years.

Round 2

Reviewer 1 Report

The manuscript has been sufficiently improved. 

Author Response

Thank you

Reviewer 2 Report

The authors have addressed the comments raised by the reviewers. Besides the tracked changes version of the manuscript, a clean copy should be uploaded to facilitate comparison with the old manuscript and to see the changes, The section on future directions, in my opinion, should have been elaborated taken into account the progress made in the last 5 years. Language correction is recommended.

Author Response

Thank you for your comments. Accordingly, we have completely revised the section "Conclusions and future direction" in light of the progress of the last 5 years, [Pg. 22-23]. As suggested, language corrections were further made, and a clean pdf copy has been uploaded.